# Functional and Genetic Characterization of Porcine Beige Adipocytes

**DOI:** 10.3390/cells11040751

**Published:** 2022-02-21

**Authors:** Lilan Zhang, Silu Hu, Chunwei Cao, Chuanhe Chen, Jiali Liu, Yu Wang, Jianfeng Liu, Jianguo Zhao, Cong Tao, Yanfang Wang

**Affiliations:** 1State Key Laboratory of Animal Nutrition, Institute of Animal Science, Chinese Academy of Agricultural Sciences, Beijing 100193, China; zhanglilancaas@163.com (L.Z.); chenchuanhe2022@126.com (C.C.); adai1013@126.com (J.L.); 2College of Animal Science and Technology, China Agricultural University, Beijing 100193, China; liujf@cau.edu.cn; 3College of Animal Science and Technology, Sichuan Agricultural University, Chengdu 611130, China; erichu121@foxmail.com; 4Guangdong Provincial Key Laboratory of Malignant Tumor Epigenetics and Gene Regulation, Guangdong-Hong Kong Joint Laboratory for RNA Medicine, Sun Yat-Sen Memorial Hospital, Sun Yat-Sen University, Guangzhou 510275, China; caochw5@mail.sysu.edu; 5State Key Laboratory of Stem Cell and Reproductive Biology, Institute of Zoology, Chinese Academy of Sciences, Beijing 100101, China; wangyu950306@163.com (Y.W.); zhaojg@ioz.ac.cn (J.Z.); 6Guangdong Laboratory for Lingnan Modern Agriculture, Guangzhou 510642, China

**Keywords:** pig, beige adipocyte, differentiation, transcriptome, ITGA2, CNN1, thermogenesis

## Abstract

Beige adipocytes are a distinct type of fat cells with a thermogenic activity that have gained substantial attention as an alternative cellular anti-obesity target in humans. These cells may provide an alternative strategy for the genetic selection of pigs with reduced fat deposition. Despite the presence of beige adipocytes in piglets, the molecular signatures of porcine beige adipocytes remain unclear. Here, white and beige adipocytes from Tibetan piglets were primarily cultured and differentiated. Compared to the white adipocytes, the beige adipocytes exhibited a stronger thermogenic capacity. RNA-sequencing-based genome-wide comparative analyses revealed distinct gene expression profiles for white and beige adipocytes. In addition, two genes, integrin alpha-2 (ITGA2) and calponin 1 (CNN1), which were specifically differentially expressed in porcine beige adipocytes, were further functionally characterized using a loss-of-function approach. Our data showed that both genes were involved in differentiation and thermogenesis of porcine beige adipocytes. Collectively, these data furthered our understanding of gene expression in porcine white and beige adipocytes. Elucidating the genetic basis of beige adipogenesis in pigs will pave the way for molecular design breeding in both pigs and large animal models of human diseases.

## 1. Introduction

Mammalian adipose tissue is an important endocrine organ regulating energy balance. Three kinds of adipocytes, white, brown, and beige (or brite), have been identified and can be classified into two categories according to their function: white adipocytes store excess energy in the form of triglycerides, and brown and beige adipocytes dissipate energy via adaptive thermogenesis. Compared to white adipocytes, which comprise unilocular lipid droplets and few mitochondria, brown adipocytes are rich in mitochondria and comprise multilocular lipid droplets [1]. The morphology of beige adipocytes is similar to brown adipocytes, exhibiting multilocular lipid droplets and dense uncoupling protein 1 (UCP1)-positive mitochondria [2].

The origins of these three adipocytes are distinct, as white adipocytes are differentiated from myogenic factor 5 (Myf5)-negative cells, while brown adipocytes share origins with skeletal muscles and are derived from Myf5-positive cells [3]. Beige adipocytes are interspersed within white adipose tissue in response to various stimuli, such as cold exposure and exercise [4]. Beige adipocytes have been demonstrated to be cellularly derived via de novo differentiation from resident precursors or via the reprogramming of mature white adipocytes [1]. Due to certain metabolic benefits, the induction and activation of beige fat have attracted considerable interest [5,6,7].

At present, the beige adipocytes of humans and mice have been molecularly characterized in vivo and in vitro [8,9,10]. Similar to brown adipocytes, the thermogenic activity of beige adipocytes depends on UCP1 activation. In addition, many marker genes specific for beige adipocytes have been identified in mice and humans, such as TBX1, TMEM26, CD137, FGF21, P2RX5, PAT2, CAR4, and CITED1 [11,12]. Moreover, many beige adipocyte-specific transcription factors (TFs), including EBF2, ZFP516, and PRDM16, have been identified and proven to play an important role in regulating beige adipocytes differentiation [13,14,15]. These markers not only deepen understanding of beige adipocyte-related biology but also provide therapeutic molecular targets for obesity and its related metabolic disorders [16].

UCP1 creates a proton leak across the inner mitochondrial membrane, diverting protons away from ATP synthesis and resulting in heat dissipation [17]. However, it is commonly accepted that pigs lack functional brown adipose tissue (BAT) and UCP1 [18], suggesting that the thermogenesis process in pigs differ from that in humans and rodents [18,19,20]. Piglets are sensitive to the cold and rely on shivering to keep them warm by huddling together. In a previous study, we observed the presence of beige adipocytes in the adipose tissues of cold-resistant pig breeds, such as Tibetan and Min pigs [21], while no beige adipocytes were found in cold-sensitive pigs (Bama and Wuzhishan) after four hours of cold stimulation [21]. The presence of beige adipocytes in piglets upon acute cold exposure was also observed [22], and beige adipogenesis can be achieved by co-overexpressing porcine PPARGC1A and mouse UCP1 in pig preadipocytes [23]. However, the understanding of beige adipocytes formation in pigs is limited, and whether the markers identified in human and mouse adipocytes can unambiguously define porcine beige adipocytes is currently unknown.

Here, stromal vascular fraction (SVF) cells from the inguinal subcutaneous adipose tissue of Tibetan piglets were differentiated into beige or white adipocytes, and the beige adipocytes were molecularly characterized. Moreover, two genes that regulate beige differentiation and thermogenic capacity were identified by using a loss-of-function approach.

## 2. Materials and Methods

### 2.1. Isolation, Culture, and Differentiation of Porcine SVF Cells

The animal study was reviewed and approved by the Animal Ethics Committee of the Institute of Animal Science, Chinese Academy of Agricultural Sciences (31 March 2020, CAAS; approval No: IAS2020-21). Inguinal subcutaneous white adipose tissues (IWAT) were collected from three 1-month-old male Tibetan piglets, respectively. Adipose tissue was dissected, washed with DPBS (HyClone), supplemented with 5% penicillin–streptomycin (P/S), minced, and digested for 60 min at 37 °C in Dulbecco’s Hanks Balanced Salt Solution (D-Hanks) (Solarbio) containing 2 mg/mL of collagenase type I (C0130, Sigma). The tissue suspension was filtered through a 70-μm cell strainer and centrifuged at 1500 r/min for 10 min to pellet the SVF cells. The cell pellet was resuspended in Dulbecco’s modified Eagle’s medium (DMEM)/F12 medium (SH30023.01, GE HyClone) supplemented with 10% fetal bovine serum (FBS) (ST30-3302, PAN) and 1% P/S and plated. A total of 1 × 10^6^ cells were plated onto 100 mm culture plates (#430167, Corning) and maintained at 37 °C in a humidified environment containing 5% CO_2_. Upon reaching 80–90% confluence, the cells were seeded in 12-well plates (2 × 10^5^ cells/well) for the following studies.

For adipogenic differentiation, pig adipose tissue-derived SVF cells were induced into fully differentiated beige or white adipocytes, as previously described [21]. Briefly, SVF cells were grown for an additional 2 days after reaching 100% confluence and then induced to differentiate with high-glucose DMEM (SH30243.01, GE HyClone) supplemented with 1% P/S. For white adipocyte differentiation, cells were treated with white adipocyte induction medium (basic differentiation medium containing 10% FBS, 20 mM HEPES (pH 7.4), 5 μg/mL of insulin, 17 μM of pantothenate, 33 μM of biotin, 1 μM of dexamethasone, 0.25 mM of isobutylmethylxanthine and 50 of mM rosiglitazone) for 5 days. Then, the cells were differentiated for 3 days in mature medium (5 μg/mL insulin, 17 μM of pantothenate, 33 μM of biotin and 1 μM of dexamethasone). For beige adipocytes differentiation, cells were treated with beige adipocyte induction medium (basic differentiation medium containing 2% FBS, 1 nM of T3, 1 μM of dexamethasone, 0.5 mM of isobutylmethylxanthine and 2 μM of rosiglitazone) and the medium was replaced every other day for 8 days. Fully differentiated adipocytes were used for the following experiments.

### 2.2. Western Blot

Beige adipocytes were lysed with M-PER^TM^ Mammalian Protein Extraction Reagent (#78503, Thermo Fisher Scientific, Waltham, MA, USA) supplemented with a cocktail of protease inhibitors (#04693159001, Roche, Indianapolis, IN, USA). The cell lysate was centrifuged at 12,000× *g* for 20 min, and the supernatant was used for the following study. Proteins (20–50 μg) were separated by 10% SDS-PAGE, and then transferred onto PVDF membranes (IPVH00010, Merck-Millipore, Madison, WI, USA). The membranes were sealed with 5% skim milk/TBST for 2 h and then incubated with primary antibodies at 4 °C overnight. The membranes were rinsed with TBST three times for 10 min each and then incubated with secondary antibodies. The protein bands were captured by a FluorChem M Fluorescent Imaging System (Tanon 5200, Tanon, Shanghai, China) with Tanon™ High-sig ECL Western blotting Substrate (#180–501, Tanon, Shanghai, China). The following primary antibodies were used: CNN1 (1:2000, #13938-1-AP, Proteintech, Rosemont, IL, USA), β-tubulin (1:2000, #2146S, Cell Signaling, Danvers, MA, USA), and PPARGC1A (1:2000, ab106814, Abcam, Cambridge, MA, USA). An anti-rabbit IgG HRP-conjugated antibody (1:2000, # 7074S, Cell Signaling, Danvers, MA, USA) and rabbit anti-goat IgG antibody (1:5000, # NHS0049, Novogene, Beijing, China) were used as secondary antibodies.

### 2.3. RNA Isolation, RT–PCR, and QPCR Analysis

Cells were washed with DPBS and lysed directly in 12-well plates with TRIzol reagent (#15596018, Thermo Fisher Scientific, Waltham, MA, USA). Total RNA was extracted according to the manufacturer’s instructions. The quality and purity of total RNA were measured by a microspectrophotometer (Nano100, NanoDrop Technologies, Wilmington, DE, USA) and an Agilent 2100 Bioanalyzer (Agilent, Santa Clara, California, USA,). One microgram of total RNA from every sample was reverse transcribed into cDNA using the PrimeScript™ RT reagent Kit with gDNA Eraser (RR047A, Takara, Tokyo, Japan). In addition to RNA sequencing (RNA-seq), total RNA was also used for gene validation by real-time PCR. The differentiated adipocytes used for RNA-Seq and qPCR were not from the same batch, while all the samples used for qPCR were from the same batch of differentiated adipocytes. Real-time PCR was performed in triplicate with TB Green^®^ Premix Ex Taq™ (RR420A, Takara, Tokyo, Japan) and quantified by a QuantStudio 3 real-time PCR instrument (Thermo Fisher Scientific, Waltham, MA, USA). The 20-μL PCR components included 2 μL (20 ng) of RT product, 10 μL TB Green Premix Ex Taq, 0.4 μL of primer (10 μM) and 0.4 uL ROX Reference Dye II. PCR conditions were 95 °C for 30 s, 40 cycles of 95 °C for 5 s and 60 °C for 34 s. The 18S rRNA gene was used as a housekeeping gene to normalize the target gene expression levels using the 2^−ΔΔCT^ method. The primer sequences are listed in Appendix A.

### 2.4. mRNA Sequencing, RNA-seq Data Analysis and Functional Analysis

Sequencing libraries and RNA-seq were conducted at Shanghai Personal Biotechnology Co., Ltd. RNA samples with high purity (OD 260/280 ≥ 2.0) and high integrity (RIN > 7) were used to construct the cDNA library. More detailed information is presented in a previous study [21]. Genes with a fold change (FC) > 1.5 and *p* < 0.05 after correcting for multiple testing were classified as differentially expressed genes (DEGs). Gene Ontology (GO) enrichment analysis of upregulated and downregulated genes was performed using the Metascape database (https://metascape.org, accessed on 16 June 2021). GO terms with *p* values < 0.05 were regarded as statistically significant. Gene set enrichment analysis (GSEA) was used to evaluate the enriched pathways in each comparative group based on E-MTAB-2602. R studio based on R version 4.1.0 and an online website (https://software.broadinstitute.org/morpheus, accessed on 16 June 2021) were used for statistical analysis and mapping.

### 2.5. Oil Red-O Staining

Cells were carefully washed twice with DPBS and fixed with 4% paraformaldehyde at room temperature for 30 min to 1 h. After washing with DPBS twice, the cells were stained with 60% saturated Oil Red-O reagent (G1260, Solarbio, Beijing, China) for 15 min and then washed with 60% isopropanol and DPBS. The Oil Red-O absorbed by cells was extracted with 100% isopropanol and detected by a SpectraMax M5 (Molecular Devices, San Jose, CA, USA) at 510 nm.

### 2.6. Fluorescence Microscopy

Cells were seeded, cultured and differentiated in glass-bottom confocal plates (D35-20-0-N, Cellvis). On the day of the experiment, 200 nM of BODIPY493/503 (D3922, Thermo Fisher Scientific) was added to the cell culture medium for 30 min followed by imaging, which was performed on a laser scanning confocal microscope (Leica TCS SP8, Wetzlar, Germany). A 63× apochromat oil-immersion mirror and AiryScan super-resolution detector were used for super-resolution imaging. The low-resolution images were taken with a 10× air objective lens [24]. All fluorophores were excited in separate orbits to avoid artifacts due to bleed-through emission. BODIPY 493/503 was excited at 488 nm.

### 2.7. Image Acquisition and Processing

Cells were imaged with a light microscope. ImageJ (V1.8.0, Rawak Software, Inc. Stuttgart, Germany) was used for image processing and normalization. Adipocyte droplet size was quantified using ImageJ software from brightfield images of individual wells of 12-well plates acquired with a 40× objective. Image contrast was enhanced, and the images were converted to 8-bit grayscale images. They were subsequently binarized, and the size and quantity of the lipid droplets were recorded.

### 2.8. Seahorse Metabolic Assays

SVF cells (1 × 10^4^) were inoculated into an XFe96 culture microplate (Seahorse Bioscience) and cultured in DMEM containing 10% FBS and 1% P/S at 37 °C in a 5% CO_2_ atmosphere, followed by the induction of differentiation. On day 6, O_2_ consumption was measured using a Seahorse Bioscience XFe96 extracellular flux analyzer. The oxygen consumption rate (OCR) was determined by sequentially adding 1.5 μM of oligomycin, 0.5 μM of carbonyl cyanide p-(trifluoromethoxy) phenylhydrazone (FCCP) and 0.5 of μM antimycin A/rotenone. The basal respiration rate, proton leak and ATP production were calculated by Wave Desktop software (Agilent, Palo Alto, California, USA). There were 10 replicates in each treatment group, and all data are shown as the mean ± standard error mean (SEM). Statistical comparisons were made by Student’s *t* test.

### 2.9. RNA Interference

Stealth RNAi™ against pig ITGA2 (si-ITGA2) and CNN1 (si-CNN1) (Jima, Shanghai, China) were used to perform RNA interference. SVF cells cultured in 12-well plates and grown to 80% confluence were transfected with siRNA with Lipofectamine RNAiMAX (#13778150, Invitrogen, Waltham, MA, USA) according to the manufacturer’s instructions. All transfection experiments were performed in triplicate. The siRNA sequences were as follows: negative control (si-NC): 5′-UUCUCCGAACGUGUCACGUTT-3′, antisense: 5′-ACGUGACACGUUCGGAGAATT-3′; si-ITGA2: 5′- GCAAGAGAUUCCGCUUAUUTT-3′, antisense: 5′- AAUAAGCGGAAUCUCUUGCTT-3′; and si-CNN1: 5′- GGUGAACGUGGGAGUGAAATT-3′, antisense: 5′- UUUCACUCCCACGUUCACCTT-3′.

### 2.10. Statistical Analysis

All experiments were performed with at least three biological replicates unless otherwise noted. For statistical analysis, GraphPad Prism 8.0.2 and JMP 10.0.0 were used. Data that were normally distributed were analyzed by a two-tailed and unpaired Student’s *t* test. Data that were not normally distributed were subjected to nonparametric analysis using the Wilcoxon/Kruskal–Wallis test. All values are expressed as the mean ± SEM. * *p* < 0.05, ** *p* < 0.01 and *** *p* < 0.001 were used as statistical significance standards.

## 3. Results

### 3.1. The Establishment of an In Vitro Model of Porcine Beige Adipogenesis

To systemically identify genes involved in porcine beige adipogenesis, we first established a protocol for beige adipocyte differentiation, as shown in Figure 1A. SVF cells were obtained from the IWAT of Tibetan pigs and then differentiated into mature white or beige adipocytes. Despite both types of adipocytes having similar differentiation abilities (Figure 1B, left panel), the sizes of the lipid droplets were distinctly different as determined by bright field microscopy analysis (Figure 1B, middle panel) and fluorescence staining (Figure 1B, right panel). To determine whether this difference could be explained by random variation or reflected the presence of truly different fat cell types, we analyzed the lipid droplet size frequencies. As shown in Figure 1C, the lipid droplets from beige adipocytes were much smaller than those from white adipocytes. To evaluate the abilities of uncoupled mitochondrial respiration and fatty acid oxidation, white and beige adipocytes were subjected to in vitro metabolic analysis using Seahorse XFe96 Analyzers. As expected, the OCR in beige adipocytes indicated that they had a significantly higher basal mitochondrial respiration rate (Figure 1D,E) and produced more ATP (Figure 1D,F) than the white adipocytes. The basal level of uncoupled respiration (proton leak) in beige adipocytes was twice as much as that in white adipocytes (Figure 1D,G). Collectively, these data suggest that beige adipocytes differ from white adipocytes in lipid droplet size and metabolism and have the metabolic ability to increase fatty acid oxidation and uncoupled respiration.

### 3.2. Distinct Transcriptional Profiles of Porcine White and Beige Adipocytes

To explore the molecular signature of porcine beige adipocytes, genome-wide RNA-seq was performed on SVF cells, white adipocytes and beige adipocytes. Principal component analysis (PCA) showed that the three groups of cells could be clearly classified (Figure 2A), suggesting distinct differences in their gene expression landscapes. Pairwise DEGs were screened by the criteria of an FC > 1.5 and a *p* value < 0.05, and volcano plots were constructed to provide a broad overview of the changes in gene expression of white/SVF (Figure 2B), beige/SVF (Figure 2C) and beige/white groups (Figure 2D). Note that the known marker genes and some candidate genes are highlighted in the volcano plots. Furthermore, GSEA showed that, compared to SVF cells, the HALLMARK_ADIPOGENESIS pathway was significantly enriched in the white adipocytes group (NES = 1.5897489, *p* = 0.0) (Figure 2E), and the GOBP_BROWN_FAT_CELL_DIFFERENTIATION pathway was significantly enriched in the beige adipocytes group (NES = 1.5934824, *p* = 0.028846154) (Figure 2F). As expected, the HALLMARK_OXIDATIVE_PHOSPHORYLATION pathway was enriched in the beige/white adipocytes comparison (NES = 1.481429, *p* = 0.078431375) (Figure 2G). In addition, the top 15 genes in each comparison group were selected for heatmap construction. Clearly, adipogenic genes, including PPARG, FABP4, ADIPQ and CEBPA, were induced in both white and beige adipocytes, while the levels of thermogenesis-related genes, such as PPARGC1A, DIO2, and EBF2, were observed to be strongly upregulated in beige adipocytes. Strikingly, mitochondrial oxidative phosphorylation pathway-related genes, such as UQCRC2, SDHB, and PDK4, showed significant induction in only beige adipocytes (Figure 2H). The molecular features of porcine white and beige adipocytes were further validated by analyzing the expression of well-recognized marker genes in humans and mice (Figure 2I).

### 3.3. Identification of Beige Adipogenesis-Related Candidate Genes

To identify porcine beige adipocyte markers, we analyzed the overlapping gene sets across each comparative group. As shown in the Venn diagram, 1228 genes (genes that were significantly upregulated in the beige/white and beige/SVF groups but nonsignificantly upregulated in the white/SVF group) were found to be specifically upregulated (Figure 3A and Appendix A), and 1376 genes (genes that were significantly downregulated in the beige/white and beige/SVF groups but nonsignificantly downregulated in the white/SVF group) were downregulated (Figure 3D and Appendix A) in porcine beige adipocytes. These genes were subjected to GO analysis, and the oxidative phosphorylation and ATP metabolic process pathways were found to be enriched among the upregulated genes (Figure 3B), while downregulated genes were found to be involved in pathways related to muscle structure development and blood vessel development (Figure 3E). In addition to the well-known beige adipocyte-related genes that we validated and showed in Figure 2I, such as PDK4, DIO2, CD137 and CIDEA, we were particularly interested in other novel beige genes. Heatmaps and qPCR analysis showed that ITGA2, AMCF-II and SEMA3A were specifically upregulated (Figure 3C,G), while CNN1, COL1A1, RIMKLB, etc., were specifically downregulated in beige adipocytes (Figure 3F,H). TFs reportedly play critical roles in cell fate decisions and differentiation, and we also screened the TFs from the DEGs list by comparison with the TFs database AnimalTFDB3 [25]. A total of 42 TFs were identified among 1288 DEGs (Figure 3I and Appendix A), including EBF2 and TAFM, which have been reported to be required for beige adipocytes commitment [26,27,28]. In addition, the downregulated genes contained 108 TFs (Figure 3J, Appendix A).

### 3.4. Comparative Analysis of DEGs in Beige Adipogenesis

Our previous studies showed that beige adipocytes were induced after cold exposure in Tibetan pigs [21]. It is of great interest to identify the key genes that regulate porcine beige adipogenesis in vitro and in vivo. We compared 4472 DEGs (up: 2076, down: 2396) from the beige/white group in the current study and 1765 DEGs (up: 862, down: 903) from a previous study (FC > 1.5, *p* < 0.05). Surprisingly, only 95 genes were found to be upregulated in both studies (Figure 4A), among which KCNK3 exhibited the largest fold change and has been recognized as a beige adipocyte marker in humans. The heatmap shows the expression profile of the top 20 genes, including KCNK3, ITGA2, PDK4, and SGK2 (Figure 4B). GO analysis showed that 95 genes were enriched in the regulation of lipid metabolic processes and cellular responses to lipids, among others (Figure 4C).

To determine whether the beige adipocyte-related genes that we identified were specific for pigs or conserved across the species, the core brown fat-selective genes conserved in mice and humans, previously reported by Shinoda et al. [10], were compared to porcine beige adipocyte-related genes (Figure 4D). The 18 overlapping genes, including previously defined beige/BAT markers, such as KCNK3, PDK4, EBF2, PDE4D and CD36, were upregulated across the three species, suggesting their conservative function in beige adipogenesis and functional maintenance. However, 14 overlapping genes were downregulated across the three species, including CYP4B1, HSPB7, TMEM38A and CPT1B, indicating that they inhibit the formation of beige adipocytes (Figure 4D,E).

### 3.5. ITAG2 and CNN1 Are Involved in Porcine Beige Adipocyte Differentiation and Thermogenesis

According to the comparative analysis described above, we found that ITGA2 was specifically and significantly upregulated in our porcine studies (Figure 4A, D). To better understand the role of ITGA2 in beige adipogenesis, porcine SVF cells were primarily cultured and transfected with si-ITGA2 or negative control (si-NC) at different time points (day −2, day 0 and day 3) during adipogenesis (Figure 5A). Significantly decreased expression levels of ITGA2 were observed in si-ITGA2-transfected cells (Figure 5B). Moreover, the expression levels of beige adipocyte marker genes (UCP3 and DIO2) and thermogenesis-related genes (PDK4 and PPARGC1A) were significantly decreased (Figure 5C). Oil Red-O staining and fluorescence imaging showed compromised beige adipocyte differentiation and lipid accumulation in si-ITGA2-transfected cells (Figure 5D,E). Next, a Seahorse assay was used to evaluate the effects of ITGA2 on uncoupled mitochondrial respiration and fatty acid oxidation in beige adipocytes, and our data showed that ITGA2 knockdown significantly reduced the basal mitochondrial respiration rate (Figure 5F,G) and proton leak (Figure 5F,H) in beige adipocytes.

In addition, the CNN1 gene was found to be specifically and significantly downregulated in porcine beige adipocytes with a high fold change. Because this gene is known as the smooth muscle lineage marker gene, it is important to further study its function in beige adipogenesis. Our data revealed decreased CNN1 expression at both the RNA and protein levels after si-CNN1 transfection (Figure 5I,J), while the expression of the thermogenesis-related genes, PDK4 and PPARGC1A, was significantly upregulated (Figure 5J, K). We found that CNN1 slightly but significantly decreased beige adipocyte differentiation (Figure 5L,M). However, in vitro metabolic analysis showed that although si-CNN1 had no effects on the basal mitochondrial respiration rate (Figure 5N,O), the proton leak was increased significantly in siCNN1-transfected cells (Figure 5N,P).

## 4. Discussion

Here, we primarily cultured functional porcine beige adipocytes from Tibetan piglets. We found that beige adipocytes differ from white adipocytes in lipid droplet size and metabolism and have the metabolic ability to increase fatty acid oxidation and uncoupled respiration. Of note, the PCA results suggested that porcine SVF cells, differentiated white adipocytes and beige adipocytes have distinct molecular profiles, as they can be clearly classified into three groups. The molecular features of porcine beige adipocytes were characterized by RNA-seq based on comparative transcriptome analyses. Porcine beige adipogenesis-related DEGs were screened, and the pathways of brown fat differentiation and oxidative phosphorylation were found to be enriched. Consistently, previously defined BAT/beige adipocyte markers in humans and rodents, including EBF2 [29], TNF receptor superfamily member 9 (CD137) [8,30], mitochondrial transcription factor A (TFAM) [31,32], PPARG coactivator 1 alpha (PPARGC1A) [33], type 2 iodothyronine deiodinase (DIO2) [34,35], and potassium channel K3 (KCNK3) [10], were indeed specifically upregulated in porcine beige adipocytes. Therefore, these well-known thermogenic genes in humans and rodents can unambiguously evaluate the thermogenic capacity of porcine adipocytes. However, it is necessary to identify pig-specific beige adipogenesis-related genes to better understand the mechanisms regulating fat deposition and energy in pigs.

In our previous in vivo study, we screened a set of potential thermogenic-related genes in cold-treated Tibetan pigs. As shown in Figure 4A, stringent porcine beige adipogenesis-related candidate genes were defined by analyzing the in vitro and in vivo datasets in combination. Accumulating evidence has revealed that beige adipocytes can be either reprogrammed from mature white adipocytes or de novo differentiated from specific resident precursor populations in subcutaneous white adipose tissues [36]. At this point, the overlapping genes, especially the TFs, might be strong candidates that regulate porcine beige de novo differentiation, although more experiments need to be conducted to support this speculation.

Correlation analyses of unknown genes and UCP1 expression have been used to identify beige adipocyte/BAT-related genes in humans. However, this approach is not applicable in pig studies due to the lack of a functional UCP1 gene in pigs [18,37]. Despite that a group of thermogenesis-related genes was shown to be conserved across the species by comparative genomic analysis, the identification of novel key beige adipocyte markers in pigs remains significant for the potential improvement of fat deposition-related economic traits.

The successful establishment of an in vitro beige adipocyte model provides materials for identifying novel beige adipocyte markers in pigs. A member of the integrin family, ITGA2, exhibited an expression pattern similar to that of known beige/brown adipocyte markers in beige adipocyte cultures, as well as in cold-treated adipose tissues from Tibetan pigs; moreover, this gene was proven to be necessary for porcine beige adipocyte differentiation and thermogenesis. ITGA2 encodes the alpha subunit of a transmembrane receptor for collagens and forms a heterodimer with a beta subunit to mediate cell adhesion [38,39]. Integrin family members have been reported to be involved in BAT-mediated thermogenesis [9]. Significant positive correlations between the expression levels of ITGA10 or ITGB1 in preadipocytes and the UCP1 levels in differentiated mature human BAT adipocytes were observed, and ITGB1 was further identified as a preadipocyte surface biomarker for the prediction of thermogenic capacity [9]. These data, together with the finding that ITGA2 was significantly induced in porcine beige adipogenesis, suggest the potential role of integrin family members in beige adipocyte differentiation and function. The detailed molecular mechanisms need to be further explored, particularly at a single-cell resolution.

The CNN1 gene is also of interest. As a smooth muscle lineage-selective genes, CNN1 was found to be significantly downregulated in human BAT adipocytes [10]. Consistently, decreased expression of CNN1 was also observed in porcine beige adipocytes. It has been reported that smooth muscle lineage-selective genes are abundantly expressed in mouse beige adipocytes, suggesting that a group of beige adipocytes arises from smooth muscle–like precursors [40]. In this context, we speculate that the suppression of CNN1 might repress smooth muscle cell differentiation and promote beige adipocyte formation. At this point, whether CNN1 is the key molecule responsible for switching the phenotypes of both cell types and involved in cell fate decisions needs to be further addressed.

In summary, our study defined the TFs and genes underlying beige adipogenesis in pigs based on the comparative transcriptome strategy. This study improves our understanding of beige adipogenesis in pigs and more importantly, constitutes a foundation for future porcine beige adipocyte studies. A full understanding of the molecular architecture underlying beige adipocyte recruitment in pigs could potentially benefit the pig breeding programs which toward to less fat deposition, but also provide valuable data for combating obesity and related metabolic diseases in humans.

## Figures and Tables

**Figure 1 cells-11-00751-f001:**
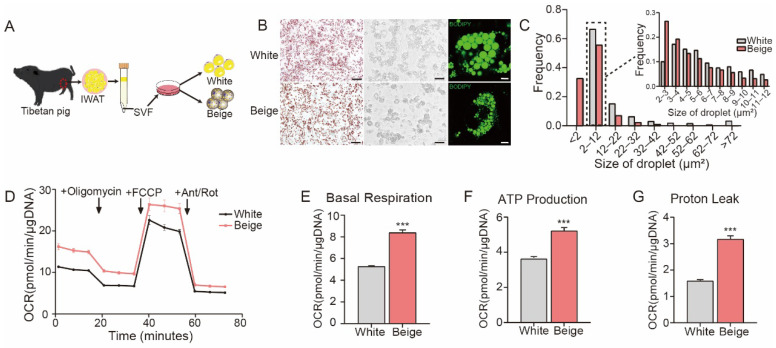
The Establishment of an in vitro model of porcine beige adipogenesis. (**A**) Schematic illustration of the in vitro differentiation of porcine beige and white adipocytes. (**B**) Oil Red-O staining (left panel), bright field imaging (middle panel), BODIPY 493/503 fluorescence staining (right panel) of white and beige adipocytes (scale bars are 200 μm, 50 μm and 10 μm, respectively). (**C**) Frequency statistical data of lipid droplet size measurements in white and beige adipocytes based on the bright field images shown in (**B**) (white: *n* = 5943; beige: *n* = 5974, pixel: 1 μm). (**D**) The oxygen consumption rates (OCRs) of white and beige adipocytes were measured by the Seahorse Mito Stress Test assay. Oligomycin (1.5 μM), FCCP (0.5 μM) and antimycin A/rotenone (0.5 μM) were added at the indicated time points. The data were normalized to the total DNA content. (**E**–**G**) The basal respiration rate (**E**), ATP production (**F**) and proton leak (**G**) were calculated based on the data presented in D (*n* = 10). All data are shown as the mean ± SEM. The *p* value was calculated by unpaired two-tailed Student’s *t* test. *** *p* < 0.001.

**Figure 2 cells-11-00751-f002:**
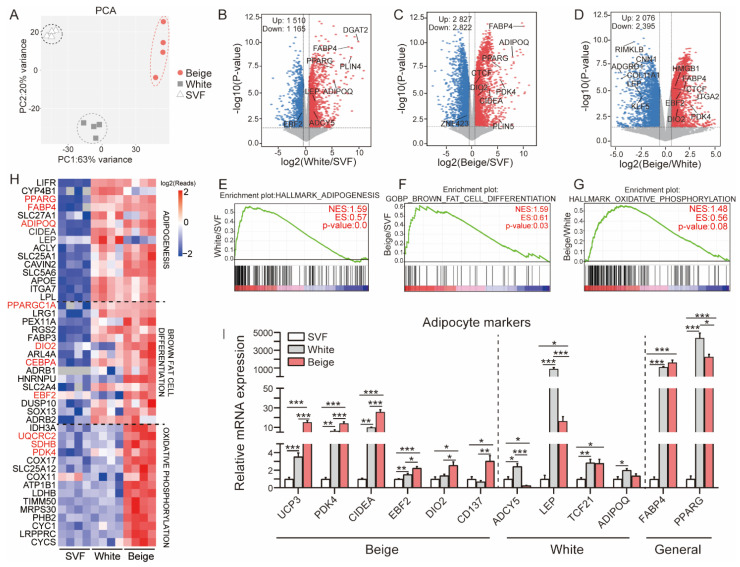
Distinct transcriptional profiles of porcine white and beige adipocytes. (**A**) Principal component analysis (PCA) plots representing SVFs, and white and beige adipocytes based on transcriptome data (*n* = 4 for each group). (**B**–**D**) Volcano plots showing a global overview of the gene expression profiles of SVF cells and white and beige adipocytes; well-known adipocyte marker genes are highlighted. Up: upregulated genes; Down: downregulated genes. (**E**–**G**) Gene set enrichment analysis (GSEA) of the RNA-seq data. GSEA showing the enriched pathways in white adipocytes compared with SVF cells (**E**). Comparison of a signature from beige adipocytes to genes upregulated in SVF cells as described; the GSEA results showed enrichment for brown fat cell differentiation (**F**). GSEA of beige adipocyte signatures compared to the list of ranked upregulated genes from white adipocytes revealed that the oxidative phosphorylation pathway was enriched (**G**). (**H**) A heatmap was constructed based on the top 15 genes involved in adipogenesis (**E**), brown fat cell differentiation and the oxidative phosphorylation (**G**) pathway. (I) Relative expression levels of adipocyte marker genes in three groups of cells (*n* = 4–6). Data are shown as the mean ± SEM. * *p* < 0.05, ** *p* < 0.01, and *** *p* < 0.001.

**Figure 3 cells-11-00751-f003:**
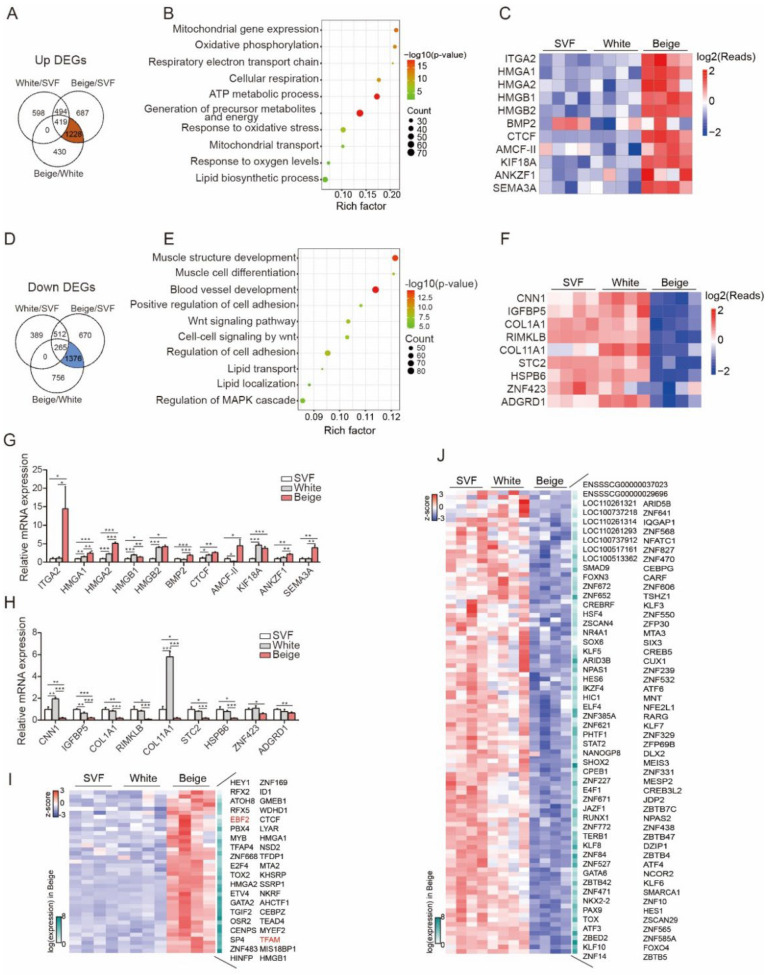
Identification of beige adipogenesis-related candidate genes. (**A**,**D**). Venn diagram showing the overlapping genes that were specifically and significantly upregulated (**A**) or downregulated (**D**) in the beige/SVF and beige/white groups (FC > 1.5, *p* < 0.05). Note that a total of 1228 upregulated and 1376 downregulated genes were included based on the criteria. (**B**,**E**). The above 1228 upregulated and 1376 downregulated genes were subjected to GO analysis, and the dot plot shows the most significantly enriched pathways. The color of the dot represents the *p* value, and the size of the dot represents the number of differentially expressed transcripts. (**C**,**F**). Heatmap of the select genes among the 1228 (**C**) and 1376 (**F**) genes. The color scale shows the z score read counts representing the mRNA level of each gene in the blue (low expression)—white–red (high expression) scheme. (**G**,**H**). The relative expression levels of the significantly upregulated genes among the 1228 genes identified (**G**) and of the significantly downregulated genes among the 1376 genes identified (**H**) in SVF cells and white and beige adipocytes were validated by qPCR (*n* = 4–6). (**I**,**J**). Heatmap of all transcription factors among the 1228 upregulated genes (**I**) and 1376 downregulated genes (**J**). All data are shown as the mean ± SEM. *p* value was calculated by the unpaired two-tailed Student’s *t* test. * *p* < 0.05, ** *p* < 0.01, and *** *p* < 0.001.

**Figure 4 cells-11-00751-f004:**
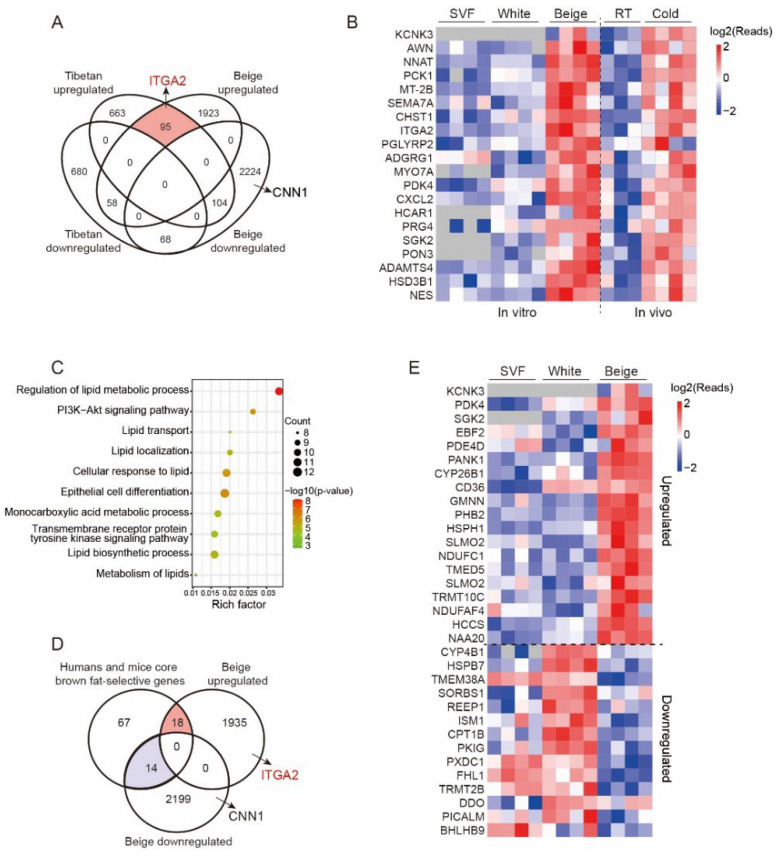
Comparative analysis of differentially expressed genes in beige adipocytes from different samples. (**A**). Venn diagram showing the overlapping genes that are specific and significant in the beige/white and Tibetan pig comparisons after cold stimulation (FC >1.5, *p* < 0.05). Note that a total of 95 upregulated genes and 68 downregulated genes were included based on the established criteria. (**B**) Expression profiles of the top 20 genes enriched in vitro and in vivo. The color scale shows the z score read counts representing the mRNA level of each gene in the blue (low expression)-white–red (high expression) scheme. RT: room temperature, cold: 4 °C for 4 h. (**C**) GO analysis of 95 upregulated genes overlapping in (**A**). The dot plot shows the most significantly enriched pathways. The color of the dot represents the *p* value, and the size of the dot represents the number of DEGs. (**D**) Venn diagram showing the beige adipocyte-related genes specific for pigs and conserved across the species (FC > 1.5, *p* < 0.05). Note that a total of 18 upregulated genes and 14 downregulated genes were conserved across humans, mice and pigs. Human and mouse core brown fat-selective genes: List of core brown fat-selective genes conserved in mice and humans. (**E**) Expression profiles of the 32 genes in (**D**) conserved across the species in the SVF, white and beige adipocytes.

**Figure 5 cells-11-00751-f005:**
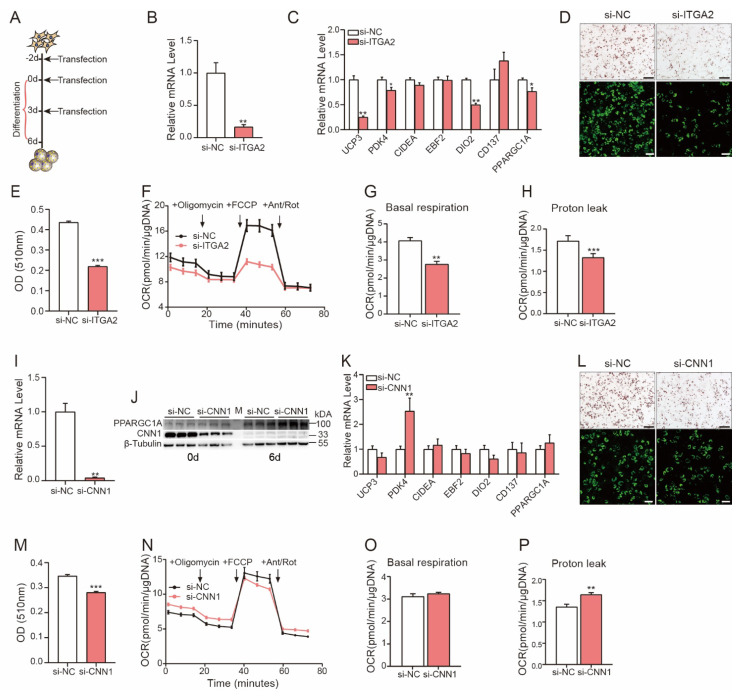
ITAG2 and CNN1 are involved in porcine beige adipocyte differentiation and thermogenesis. (**A**) Schematic of si-ITGA2 or si-CNN1 treatment during adipogenic induction. (**B**) The knockdown efficiency of si-ITGA2 was examined by qPCR on day 6. (**C**) Expression of beige adipocyte markers in cells transfected with si-ITGA2. (**D**) Effect of si-ITGA2 transfection on beige adipogenesis, as determined by Oil Red-O staining (top, scale bar is 200 μm) and fluorescence staining (bottom, scale bar is 100 μm) on day 6. (**E**) Quantitative analysis of the Oil Red-O staining data shown in D via OD measurements. F-H. OCR in differentiated porcine beige adipocytes transfected with si-NC or si-ITGA2 (**F**); the basal cellular respiration rate (**G**) and proton leak (**H**) were calculated (*n* = 10). (**I**,**J**) The knockdown efficiency of si-CNN1 was examined by qPCR and Western blotting on day 6. (**K**) Expression of beige adipocyte markers in cells transfected with si-CNN1. (**L**). Effect of si-CNN1 transfection on beige adipogenesis, as determined by Oil Red-O staining (top, scale bar is 200 μm) and fluorescence staining (bottom, scale bar is 100 μm) on day 6. (**M**) Quantitative analysis of the Oil Red-O staining data shown in L via OD measurements. (**N**–**P**) OCR in differentiated porcine beige adipocytes transfected with si-NC or si-CNN1 (**N**); the basal cellular respiration rate (**O**) and proton leak (**P**) were calculated (*n* = 10). The Seahorse assay data were normalized to the total DNA content. All data are shown as the mean ± SEM. *P* values were calculated by the unpaired two-tailed Student’s *t* test. * *p* < 0.05, ** *p* < 0.01, *** *p* < 0.001.

## Data Availability

The RNA-seq raw data are available in the Genome Sequence Archive in BIG Data Center, Beijing Institute of Genomics (BIG), Chinese Academy of Sciences, under accession number CRA005327 (http://bigd.big.ac.cn/gsa, released on 21 December 2021).

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
