# Peer review of "Functional and Genetic Characterization of Porcine Beige Adipocytes"

_cells, 2022, doi:10.3390/cells11040751_

Round 1

Reviewer 1 Report

Cells

Summary: The manuscript of Zhang et al. “ITGA2 is indispensable for porcine beige adipogenesis” studies the molecular signature of beige adipocyte tissue in pigs. The authors identified differentially expressed genes compared to white adipose tissue that could represent marker or target specific for beige adipose tissue in pigs. In particular ITGA2 seems important for the beige adipocyte differentiation.

General comments:

The topic of the manuscript is original and relevant in the field, but the manuscript need to be English edited, Introduction should be revised, Maretials and Methods should give more information, Statistical analyses should be revised.

English need a moderate editing. Very often the subject of the sentence is missing or is difficult to understand.

Specific comments

Abstract:

Abstract should be completely revised. The mention to previous study should be moved to introduction and the beige adipose tissue should be introduced better.

Line 19: the sentence is not clear

Line 28: the sentence is not clear

Introduction:

Before all the “[“ please put a space.

The introduction is too short lacks a deep description of the 3 types of adipocytes: white, brown and beige (origins, characteristics, functions).

Line 46-51: to which species the sentence is referred to?

Line 52: beige “adipocyte” or “fat”

Line 53: please add the reference

Line 54: the role of UPC1 should be better described

Line 61: In what kind of pigs? Which is the difference with the previous sentence?

Line 62: In vitro? Where? Please clarify

Materials and Methods:

Line 73: the authors should indicate the number of authorization and the date. How many animals were used?

Line 75: the samples were obtained at a slaughter house?

Line 82: the cells were plated in flask, petri dishes, 6-weel plates…..? At which concentration were plated the cells?

Line 96: did the authors checked the cells at FACS? How they check the purity of the cells? There is fibroblast (or other cell) contamination?

Line 100: was the cell lysate centrifuged and the pellet discarded? How many ug of total proteins were loaded per lane in WB?

Line 102 and 104: please mention the secondary antibodies used

Line 107: the primary antibodies were used together on the same membrane? The authors used replicates of membranes for the different antibodies? Did they strip the membranes?

Line 115: please detail the concentration of the Real Time PCR reagents. Did the authors use random primers for the retro transcription? Please detail the amplification protocol (how many cycles, temperatures….). Did the authors validate the primers? Which sample was used as calibrator for the formula 2-DDCt?

Line 153: use “were” instead of “are”

Line 164: the first time all the

Line 171: use “were” instead of “was”

Line 180: add “in” after “at least”; add a space between “replicates” and “unless”

Line 184: before use a parametric test, the authors should check the type of data distribution by a normality test.

Results:

Line 189: an approach for what?

Line 196: the dimension difference is significant?

Line 203-205: this sentence should be moved to Discussion section

Line 231-240: this sentence should be moved to Discussion section

Line 257: compared to white?

Line 267: compared to white?

Line 289: it is not clear why the authors mention in vivo experiment….

Line 315: please delete the “.” After “CD36”

Line 328: why the authors did not check the marker gene expression?

Line 332: did the authors make a WB to check the silencing of ITGA2?

Why the authors did not use the FACS analysis to study the different markers?

Discussion:

Line 365-366: the sentence is not clear

Author Response

General comments:

The topic of the manuscript is original and relevant in the field, but the manuscript need to be English edited, Introduction should be revised, Maretials and Methods should give more information, Statistical analyses should be revised.

English need a moderate editing. Very often the subject of the sentence is missing or is difficult to understand.

Response: Thank you for these valuable suggestions. We revised the introduction section and provided more information about the Materials and Methods. In addition, our manuscript has been polished by native English language speakers. Hopefully, the reviewer will be satisfied with the revised version of our manuscript.

Specific comments

Abstract:

Abstract should be completely revised. The mention to previous study should be moved to introduction and the beige adipose tissue should be introduced better.

Response: Thank you for this suggestion. We provided more information on beige adipose tissue (lines 16-17) and revised the description of the previous study in the Abstract (lines 19-20).

Line 19: the sentence is not clear

Response: Thank you for this comment. We revised this sentence (Lines 18-19).

Line 28: the sentence is not clear

Response: Thank you for this comment. In this sentence, “identify a novel porcine beige marker” is kind of overstatement, and we revised this statement in accordance with this suggestion (Line 25-28).

--Introduction:

Before all the “[“ please put a space.

Response: This mistake has been revised.

The introduction is too short lacks a deep description of the 3 types of adipocytes: white, brown and beige (origins, characteristics, functions).

Response: Thank you for this suggestion. We added the related information (Lines 34-44).

Line 46-51: to which species the sentence is referred to?

Response: We added “in mice and humans” on Line 53.

Line 52: beige “adipocyte” or “fat”

Response: We added “adipocyte” on Line 54.

Line 53: please add the reference

Response: The reference has been added (Line 59).

Line 54: the role of UPC1 should be better described

Response: We added “UCP1 creates a proton leak across the inner mitochondrial membrane, diverting pro-tons away from ATP synthesis and resulting in heat dissipation [17]” (Line 60-61).

Line 61: In what kind of pigs? Which is the difference with the previous sentence?

Response: Actually, we didn’t find the pig breed information in that reference. After personal communication, we knew that they used the Yorkshire pig breed. So, we added this information on Line 69. The difference with the previous sentence is that they observed the beige formation in different pig breeds.

Line 62: In vitro? Where? Please clarify

Response: Yes, this study was performed in vitro. We added “in pig preadipocytes” on Line 70. 

Materials and Methods:

Line 73: the authors should indicate the number of authorization and the date. How many animals were used?

Response: We added the number and date of authorization on Lines 82-83. We used three piglets for SVF cell culture. We updated the information on Line 84.

Line 75: the samples were obtained at a slaughter house?

Response: We purchased the piglets from a pig farm and isolated adipose tissues in the laboratory.

Line 82: the cells were plated in flask, petri dishes, 6-weel plates…..? At which concentration were plated the cells?

Response: A total of 1´106 cells were plated in 100 mm petri dishes. Upon reaching 80-90% confluence, the cells were seeded in 12-well plates (2´105 cells/well). This information was added on Lines 91-94. In addition, we added the information related to the Seahorse assays (Line 177).

Line 96: did the authors checked the cells at FACS? How they check the purity of the cells? There is fibroblast (or other cell) contamination?

Response: This is a great question. Yes, the cells we cultured were SVFs, a mixture of preadipocytes, mesenchymal stem cells and fibroblasts. We did not check the purity of the cells or isolate the preadipocytes by FACS. Based on the Oil Red-O staining and marker gene expression, the differentiation efficiency was sufficiently high for our study.   

Line 100: was the cell lysate centrifuged and the pellet discarded? How many ug of total proteins were loaded per lane in WB?

Response: Cell lysate was centrifuged at 12,000 g for 20 min, and the supernatant was used for the following studies. A total of 20-50 μg of total protein was loaded per lane in the WB (Lines 112-113).

Line 102 and 104: please mention the secondary antibodies used

Response: We added “An anti-rabbit IgG HRP-conjugated antibody (1:2000, # 7074S, Cell Signaling) and rabbit anti-goat IgG antibody (1:5000, # NHS0049, Novogene) were used as secondary antibodies.” on Lines 121-123.

Line 107: the primary antibodies were used together on the same membrane? The authors used replicates of membranes for the different antibodies? Did they strip the membranes?

Response: the primary antibodies were used together on the same membrane, and we cut the membrane based on the molecular weight of the target proteins. We did not strip the membranes.

Line 115: please detail the concentration of the Real Time PCR reagents. Did the authors use random primers for the retro transcription? Please detail the amplification protocol (how many cycles, temperatures….). Did the authors validate the primers? Which sample was used as calibrator for the formula 2-DDCt?

Response: We updated the detailed real-time PCR protocols, including the reagent information and the amplification protocol (Lines 134-138), in the revised manuscript. For the retro transcription, we used mixed RT primers, including Random 6 mers and Oligo dT Primer. We did validate the primers, and none of the primer pairs had nonspecific amplification. The 18S rRNA gene was used as a housekeeping gene to normalize the target gene expression data by the 2−ΔΔCT method.

Line 153: use “were” instead of “are”

Response: This mistake has been revised.

Line 164: the first time all the

Response: We added the full term of OCR and FCCP (on Line 180-182).

Line 171: use “were” instead of “was”

Response: This mistake has been revised.  

Line 180: add “in” after “at least”; add a space between “replicates” and “unless”

Response: This mistake has been revised.

Line 184: before use a parametric test, the authors should check the type of data distribution by a normality test.

Response: Yes, before use a parametric test, we check our data, and the data followed a normal distribution. We added “For statistical analysis, GraphPad Prism 8.0.2 and JMP 10.0.0 was used. Data that were normally distributed were analyzed by two-tailed and unpaired Student’s t test. Data that were not normally distributed were subjected to nonparametric analysis using the Wilcoxon/Kruskal-Wallis test” in Line 199-202.

Results:

Line 189: an approach for what?

Response: We added “established a protocol for beige adipocyte differentiation” on Line 207.

Line 196: the dimension difference is significant?

Response: The aim of Figure 1C was to compare the frequencies of lipid droplet size between beige and white adipocytes; statistical analysis here is not necessary. 

Line 203-205: this sentence should be moved to Discussion section

Response: we revised the manuscript based on this suggestion. (Lines 383-385).

Line 231-240: this sentence should be moved to Discussion section

Response: The sentences from Lines 231-240 are the description of Figures 2H and 2I, and we think it is acceptable to keep them in the Results section (Lines 251-260).

Line 257: compared to white?

Response: The 1,228 upregulated genes were those that were significantly upregulated in both the beige/white and beige/SVF groups but were not significantly different in the white/SVF group. We revised the manuscript accordingly (lines 275-277).

Line 267: compared to white?

Response: The 1,376 downregulated genes were those that were significantly downregulated in both the beige/white and beige/SVF groups but were not significantly different in the white/SVF group. We revised the text (lines 278-280).

Line 289: it is not clear why the authors mention in vivo experiment….

Response: Our previous in vivo study screened a set of potential thermogenic genes from cold-treated Tibetan pigs. It is of interest to screen the overlapping genes from both in vivo and in vitro studies for further study. That is the reason we performed the combination analysis.

Line 315: please delete the “.” After “CD36”

Response: This mistake has been revised.

Line 328: why the authors did not check the marker gene expression?

Response: Thanks for this question. We measured the expression levels of marker genes by QPCR. We included the results in Figure 5C and 5K, and added the related descriptions in Results sections (Lines 348-350; Line 375-376).

Line 332: did the authors make a WB to check the silencing of ITGA2?

Why the authors did not use the FACS analysis to study the different markers?

Response: These are also good questions. We used two antibodies against ITGA2 (A7629, Abclonal and       PA547193, Invitrogen) for WB analysis; unfortunately, we did not obtain good signals. Using FACS analysis to study the different markers is also a great idea; however, we failed to use FACS to obtain ITGA2+, and we will try again with other markers in the future. 

Discussion:

Line 365-366: the sentence is not clear

Response: We revised the sentence (Lines 395-398).

Reviewer 2 Report

This is an interesting and well-designed study by Zhan et al. The data is presented in a reader-friendly format and interactive. I have a few comments about the presented data and the conclusions drawn from the data.

Major:

1- The authors have compared SVF, white and beige adipocytes for the differential gene expression. The heat map presented in figure 2 H show a relatively weaker or no upregulation in adipogenesis-related genes CEBPA, FABP4, and ADIPOQ expression in the white adipocytes group. FABP4 also stands out to be a distinct gene in Figure 2 D in the beige adipocyte group. While the qRTPCR data in fig 2I indicates almost equal expression in both adipocytes groups. Considering these data sets it seems that the white adipogenesis efficiency was not adequate in the RNA sequencing experiments. The authors need to explain this discrepancy in the data.

2- The authors have concluded that ITGA2 is important for beige adipogenesis. In the absence of gene knockdown studies in the settings of white adipogenesis, this conclusion is an overstatement.

Minor:

The mention of the number of days adipogenesis was conducted before analyses will be helpful.

Author Response

Major:

1- The authors have compared SVF, white and beige adipocytes for the differential gene expression. The heat map presented in figure 2 H show a relatively weaker or no upregulation in adipogenesis-related genes CEBPA, FABP4, and ADIPOQ expression in the white adipocytes group. FABP4 also stands out to be a distinct gene in Figure 2 D in the beige adipocyte group. While the qRTPCR data in fig 2I indicates almost equal expression in both adipocytes groups. Considering these data sets it seems that the white adipogenesis efficiency was not adequate in the RNA sequencing experiments. The authors need to explain this discrepancy in the data.

Response: Thank you for these important points. (1) As shown in Figure 2B, the expression levels of FABP4 and ADIPOQ were significantly higher in white adipocytes than in SVF cells, suggesting adequate white adipogenesis efficiency. In addition, the Oil Red O staining of the cells in Figure 1B (upper panel) also indicated successful white adipocyte differentiation. The problem of Figure 2H is that we constructed the heatmap based on the original reads count. When we used the log-transformed reads count to rebuild the heatmap, the discrepancy in the data disappeared. We replaced Figure 2H in the revised version of the manuscript. (2) With regard to the expression of FABP4, we must say that the differentiated adipocytes used for RNA-Seq (Figure 2D) and qRT–PCR (Figure 2I) were not from the same batch. Despite this, the expression trend of FABP4 was the same, and a higher expression level was observed in beige adipocytes in both assays. However, the qPCR data did not reach statistical significance (fold change of beige/white was 1.5, P=0.1). Taken together, the data we showed in Figure2 is adequate to characterize the features of white and beige adipocytes.       

2- The authors have concluded that ITGA2 is important for beige adipogenesis. In the absence of gene knockdown studies in the settings of white adipogenesis, this conclusion is an overstatement.

Response: This is a great comment. We found that the ITGA2 gene was specifically upregulated in beige adipocytes, and we thus focused on the role of only the ITGA2 gene in beige adipogenesis. We agree with the reviewer that the conclusion of our manuscript is not that stringent. We revised the manuscript accordingly, including the title of the article and the abstract (lines 25-27).

Minor:

The mention of the number of days adipogenesis was conducted before analyses will be helpful.

Response: Thank you for this suggestion. We added the related description to the Materials and Methods section on Line 102-107. 

Round 2

Reviewer 2 Report

Kindly clarify which samples were used for the validation of data shown in figure 3 G and H. Is this the same material used for RNA sequencing? or is this the one used for figure 2 I validation.

Author Response

Response to Reviewer 2 comments

Kindly clarify which samples were used for the validation of data shown in figure 3 G and H. Is this the same material used for RNA sequencing? or is this the one used for figure 2 I validation.

Response: This is a great comment. The samples used for the validation of data shown in figure 3 G and H were not the same material used for RNA sequencing. These were the one used for figure 2 I validation. We added “The differentiated adipocytes used for RNA-Seq and qPCR were not from the same batch, while all the samples used for qPCR were from the same batch of differentiated adipocytes”. ( Line 133-135)

Round 3

Reviewer 2 Report

Accept